# The Use of Graph Theory for Modeling and Analyzing the Structure of a Complex System, with the Example of an Industrial Grain Drying Line

Ryszard Myhan [ID], Ewelina Jachimczyk [ID] and Marek Markowski *

Faculty of Technical Sciences, University of Warmia and Mazury in Olsztyn, ul. Oczapowskiego 2, 10-719 Olsztyn, Poland; ryszard.myhan@uwm.edu.pl (R.M.); ewelina.jachimczyk@uwm.edu.pl (E.J.)
* Correspondence: marek@uwm.edu.pl

**Abstract:** This article describes a method for analyzing and modeling a complex agrotechnological system using the example of an industrial grain drying line. Elements of graph theory were used to develop an effective tool for modeling such a system and to formally validate its structure. The proposed method can be applied to transform a general structural model into a set of relational models, to formally evaluate the resulting models' functionality, and to comprehensively analyze different variants of the process. The method can be deployed at the stage of designing and operating an industrial grain drying line, and it can also be adapted for use in other areas, such as processing lines in the agri-food industry.

**Keywords:** complex system; grain drying line; relational model; graph theory





## 1. Introduction

The concept of a system has numerous definitions, despite the fact that it is one of the fundamental tenets of contemporary science [1]. Many general classes of system have been proposed, including social activity systems. These dynamic systems involve complex interactions between humans, machines and the environment, and they serve specific goals [2,3]. Processes, in particular technological processes, are a subclass of this category of system [4]. Most processes are linear, with a distinct input and output state, and they have a temporal structure, where the beginning, duration and end of the process are clearly marked in time. However, devices that are used in linear processes generally have a complex structure, and they include subsystems with serial, parallel and feedback connections [5]. The structure of some processes can be modified by automatic control systems or the decisions made by humans (operators). These structures are generally hierarchical, and complex tasks involve interactions between numerous devices that perform different functions. Hierarchical systems have numerous decision centers, some of which directly influence control variables, whereas others merely define tasks and coordinate the operations of lower-level centers [6]. Hierarchical systems exist in all areas of human activity [7], including in agricultural production and the agri-food processing industry [8]. The latter categories of systems are highly unique because the attributes of the end product are significantly determined by the quality of raw materials and environmental conditions [9].

This article focuses on processes that take place in industrial grain drying lines. Grain processing lines are composed of dedicated machines and equipment, and despite the fact that line components are repeatable, each line constitutes an individual and unique technical solution [10,11]. The diversity of the applied technical solutions is determined by variations in a system's operating parameters that are driven by changes in the system's environment [12]. Such variations are observed in the attributes of plant materials (species-specific traits of seeds, grain moisture content, grain contamination, quantity of grain, frequency and structure of supplies), weather conditions (temperature and humidity),

delivery conditions and constraints that are largely associated with the production profile and market requirements [13]. The following questions should be answered to effectively control the operation of a grain drying line:

1. What types of treatment are required to obtain an end product with desirable attributes?
2. What are the optimal process parameters for a selected algorithm of a technological process?
3. Are raw materials of sufficiently high quality to obtain an end product with desirable attributes?
4. Can raw materials can be used in a different and more productive manner?
5. If numerous process algorithms are available, which algorithm should be applied?
6. In what way will present decisions affect future process performance?
7. Which long-term strategy will be more effective in achieving the defined objectives?

Only the first and the second question have been addressed in the literature [14–17]. However, most studies have focused only on controlling drying parameters [18–30]. This is because the answers to the above questions cannot be considered separately, and each answer should also address the remaining questions. In addition, the decisions made in every stage of the process have to account for the variations in internal and external conditions.

The above questions can be effectively addressed in operations research involving mathematical models that account for the main components of an industrial grain drying line [31–36]. Models of specific production processes and commercial steady-state process simulators (such as ProSimPlu and VMGSim) have been extensively researched, but models of alternative methods of managing raw materials that account for the structure and state of the processing line as well as external conditions have not been proposed in the literature to date. Such an attempt has been made by Jachimczyk and Myhan [37], but the described method was developed specifically for production lines in dairy plants. The proposed method also has a number of limitations, and it does not account for recurring processes during grain drying or the duration of each production process. Therefore, the aim of the present study was to expand the existing knowledge and propose cohesive methods for modeling grain drying processes.

The aim of this study was to propose a method for modeling the structure of an industrial grain drying line which could support decision-making for optimized management of grain drying. The proposed method was implemented in a programming environment (with the use of mathematical models of technological processes) to simulate multiple drying processes. The results of simulations can be applied to effectively support decision-making in the process of selecting optimal strategies for raw material handling. They can also be used at the stage of designing an industrial grain drying line.

It was assumed that the proposed method should facilitate the following:

1. The development of models describing various types of equipment and the structure of an industrial grain drying line;
2. A formal validation of the modeled structure;
3. The transformation of a general structural model into a set of relational models (alternative configurations of devices in process lines);
4. A formal validation of relational models.

The method should also incorporate elements of the decision-making system.

## 2. Analysis and Modeling of the Structure of an Industrial Grain Drying Line

### 2.1. Analysis of Technical Equipment

The analysis involved industrial grain drying lines in north-eastern Poland [38]. Data were collected from grain processing facilities to characterize the existing equipment and determine the internal structure of grain drying lines. Information was collected from more than 30 grain processing facilities, but complete data could not be obtained in all cases, and the analysis was ultimately narrowed down to 21 facilities. The components of grain drying lines were divided into seven groups based on their functions:

1. Input devices that deliver grain to the processing line;
2. Cleaning devices that remove impurities;
3. Internal transport devices, including chain, bucket, belt and screw conveyors;
4. Silos for storing grain before, during and after subsequent stages of drying;
5. Batch dryers and continuous flow dryers that are the key components of every grain drying line;
6. Control devices for controlling grain flow, configuring other line devices and, in some cases, controlling the volume of the grain stream;
7. Output devices for handling dried grain, including grain packing machines and truck loading silos.

### 2.2. Components of the Structural Model

A structural model of a grain drying line comprises various devices that are connected by streams. Each stream represents the flow of materials between devices in an empirical system. In view of the purpose and scope of the developed model, four types of streams were identified: grain dry matter stream, impurities stream, water stream and air stream. This functional classification accounts for two main subprocesses in a grain drying line, i.e., cleaning, during which a stream of impurities is removed from processed grain, and drying, during which a stream of water is removed from moist grain (Figure 1). In most cases, impurities and water are evacuated in a stream of air.

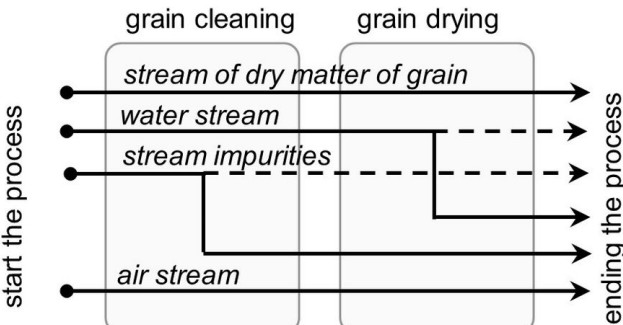

**Figure 1.** Classification of streams during grain cleaning and drying subprocesses.

In addition to the key subprocesses, subprocesses relating to transport and storage can also be identified in the technological process. The flow and division of streams in these subprocesses are presented in Figure 2. The importance of transport and storage subprocesses is determined by the extent to which they affect the properties of the processed plant material. The modeled objects were divided into the following classes based on the types of devices that are involved in each subprocess: separators, dryers, transport and storage. Two general classes of objects—dividers and environment—were additionally identified. Dividers are objects that represent valves and separating devices in the model. These objects enable temporary configuration of other class objects into process lines. The environment class was introduced to model the grain drying line as an independent system. The environment class describes the type, structure and frequency of grain deliveries as well as the properties of the delivered plant material. It also describes the parameters at the output of the grain processing line, as well as the parameters of the air stream that directly influence drying and storage subprocesses.

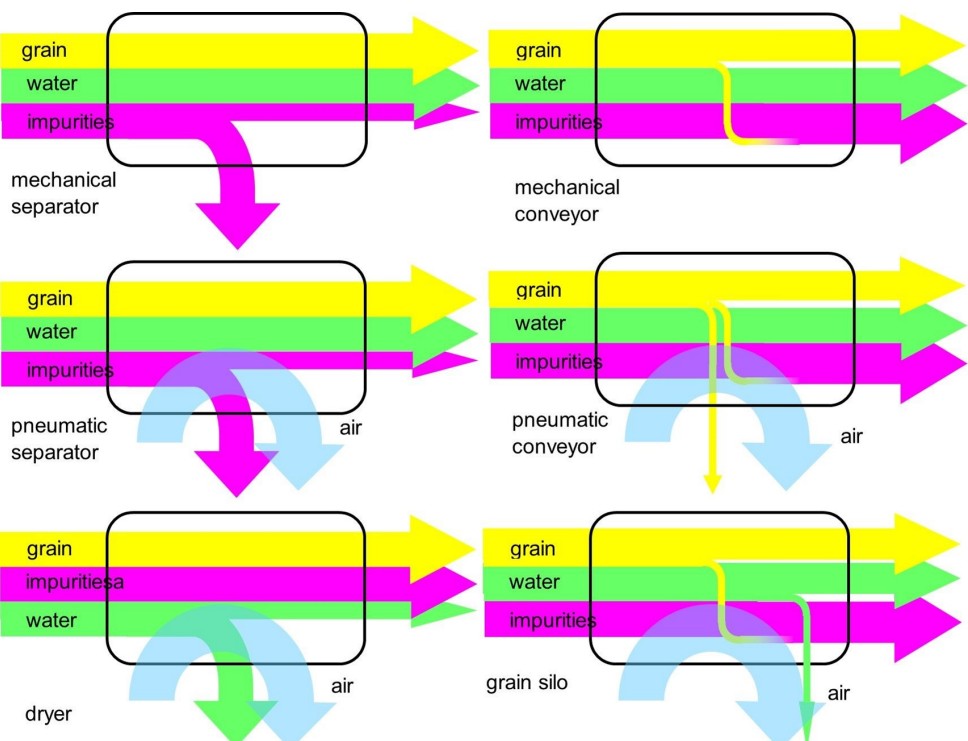

**Figure 2.** Flow and division of streams in subprocesses.

### 2.3. Limitations of the Structural Model

In the modeled structure, relationships can be established only between objects that are connected by at least one stream, and the analyzed structure has to satisfy the following requirements:

- Only one stream belonging to a given class can be present at an object's input;
- Combined streams belonging to different classes can be divided only inside the object as a result of a given subprocess;
- Streams that belong to different classes and have a common beginning and end are analyzed jointly (for example, the grain dry matter stream, water stream and impurities stream represent moist and contaminated grain);
- Objects that are not connected with other objects via the grain dry matter stream or the air stream do not belong to the structural model (which implies that the corresponding devices cannot participate in the technological process);
- A group of objects connected by streams of a given class form a process line that can perform the described process only when the first and last link in the chain is represented by an object belonging to the environment or storage class.

### 2.4. Graphic Representation of the Structural Model

A structural model was developed for a simplified hypothetical grain drying line presented in Figure 3. The line features a cleaning device, a grain dryer and two grain silos. To simplify the analyzed model, transport devices are not included in the presented diagram.

The modeled structure can be presented with the use of a simple directed graph (without loops) $G = V, U$ [39–41]. In the graph, the set of graph vertices $V$ corresponds to the set of objects $O$:

$$V = O = \{o_1, o_2, \ldots, o_n\} \qquad (1)$$

where $n$ is the number of objects in the set, and the set of arcs $U$ corresponds to the set of streams $S$:

$$U = S = S_m \cup S_z \cup S_w \cup S_p \qquad (2)$$

where $S_m$—subset of grain dry matter streams, $S_z$—subset of impurities streams, $S_w$—subset of water streams, $S_p$—subset of air streams.

These subsets are pairwise disjoint (which implies that $S_m \cap S_z \cap S_w \cap S_p = \varnothing$), and the maximum size of the arcs set is $4m$, where $m$ is the size of the largest subset:

$$m = max(\overline{\overline{S}}_m, \overline{\overline{S}}_z, \overline{\overline{S}}_w, \overline{\overline{S}}_p). \tag{3}$$

An arc directed towards a given vertex was assigned a number corresponding to the number of that vertex. When more than one arc representing the same class of streams was directed to the same vertex, all arcs were assigned the same number (Figure 4) based on the previously formulated principle that only one stream belonging to a given class can be present at an object's input. If a higher number of streams was declared in the structure, only one of these arcs can be present in the model in a given moment. The number of relational models that can be derived from the modeled structure is given by the indegree of the vertex.

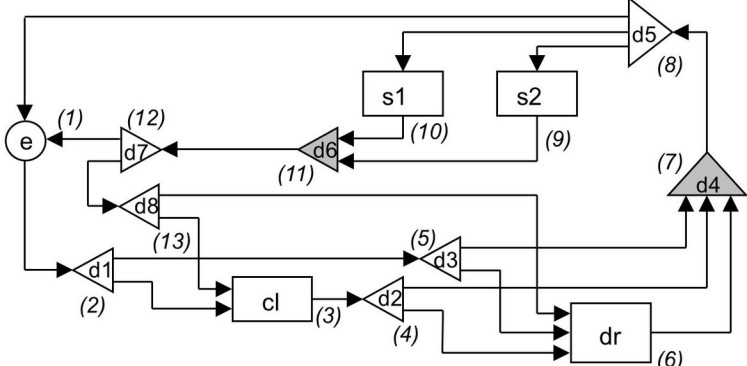

**Figure 3.** Diagram of a hypothetical grain drying line (e—environment; cl.—cleaner, dr.—dryer; s1, s2—storage, d1 ÷ d8—divider (valve), (1–13)—object number).

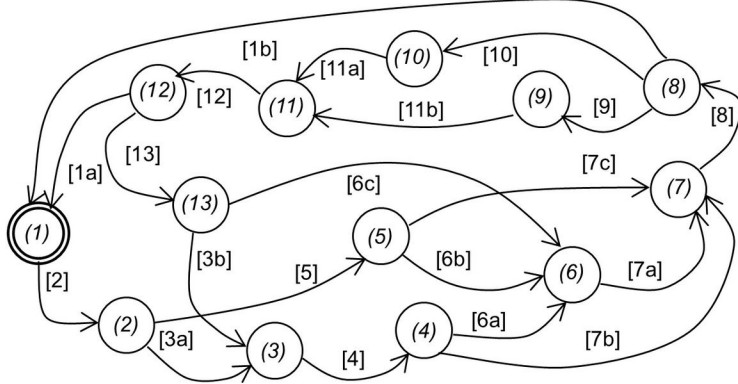

**Figure 4.** Directed graph presenting the structure of the hypothetical grain drying line depicted in Figure 3.

This graph can be unambiguously defined with the use of a three-dimensional incidence matrix [42,43]:

$$A(G) = \left[ a_{i,j,k} \right]_{n \times m \times 4}. \tag{4}$$

where $i$—vertex index, $j$—edge index, $k$—stream class index.

Element $a_{i,j,k}$ in the matrix can take on the following values:

- $a_{i,j,k} = 0$ when the $j$-th arc representing the $k$-th class of streams is not incident to the $i$-th vertex;

- $a_{i,j,k} = 1$ when the $i$-th vertex is the end of the $j$-th arc representing the $k$-th class of streams;
- $a_{i,j,k} = -1$ when the $i$-th vertex is the beginning of the $j$-th arc representing the $k$-th class of streams.

### 2.5. Formal Validation of the Graphic Representation of the Model

Errors resulting from incorrect modeling or omission of certain structural elements can emerge at the stage of modeling the structure of a processing line. Many of these errors can be identified by analyzing the incidence matrix (Equation (4)) of the structural model. For a correctly formulated model, the incidence matrix should satisfy the following formal requirements:

- If there exists vertex $o_i$ where $1 \leq i \leq n$, such that for every $1 \leq j \leq m$ and every $1 \leq k \leq 4$ $a_{i,j,k} = 0$ (i.e., the vertex $o_i$ is isolated), then this object does not belong to the model structure;
- Vertex $o_i$ represents a correctly modeled object (Table 1), if for $k = 1$, there exists $1 \leq j \leq m$, where $a_{i,j,1} = 1$, and:
  - $\sum_{j=1}^{m} a_{i,j,1} = 0$ (i.e., the vertex $o_i$ has one incoming arc and one outgoing arc) when vertex $o_i$ represents an object that does not belong to the dividers class;
  - $\sum_{j=1}^{m} a_{i,j,1} = 1 - r$ (i.e., the vertex $o_i$ has one incoming arc and at least two outgoing arcs) when vertex $o_i$ represents an object that belongs to the dividers class ($r$—number of possible connections at divider output);
- The structural model correctly depicts the analyzed process line when for every $1 \leq j \leq m$, there exist values of $k$, $i_1$ and $i_2$ where $i_1 \neq i_2$, $a_{i1,j,k} = -1$ and $a_{i2,j,k} = 1$ (i.e., the vertices $o_{i1}$ and $o_{i2}$ are connected by an arc $j$);
- The structure of the modeled line supports the grain separation subprocess if there exists a value of $i$, where $k = 1$ (grain stream) and $k = 2$ (impurities stream) satisfy the following conditions: $\sum_{j=1}^{m} a_{i,j,1} = 0$ and $\sum_{j=1}^{m} a_{i,j,2} = -1$ (i.e., the vertex $o_i$ for $k = 2$ has one incoming arc and two outgoing arcs);
- The structure of the modeled line supports the grain drying subprocess if there exists a value of $i$, where $k = 1$ (grain stream), $k = 3$ (water stream) and $k = 4$ (air stream) satisfy the following conditions: $\sum_{j=1}^{m} a_{i,j,1} = 0$, $\sum_{j=1}^{m} a_{i,j,3} = -1$ and $\sum_{j=1}^{m} a_{i,j,4} = 0$.

**Table 1.** Fragment of the incidence matrix for the diagram of the line depicted in Figure 3.

| Vertex | Grain Dry Matter Stream (k = 1) | | | | | | | | | | | | | $\Sigma$ | Object |
| | 1 | 2 | 3 | 4 | 5 | 6 | 7 | 8 | 9 | 10 | 11 | 12 | 13 | | |
|---|---|---|---|---|---|---|---|---|---|---|---|---|---|---|---|
| 1 | 1 | −1 | | | | | | | | | | | | 0 | Environment |
| 2 | | 1 | −1 | | −1 | | | | | | | | | −1 | Divider (selection at output) |
| 3 | | | 1 | −1 | | | | | | | | | | 0 | Cleaner |
| 4 | | | | 1 | | −1 | −1 | | | | | | | −1 | Divider (selection at output) |
| 5 | | | | | 1 | −1 | −1 | | | | | | | −1 | Divider (selection at output) |
| 6 | | | | | | 1 | −1 | | | | | | | 0 | Dryer |
| 7 | | | | | | | 1 | −1 | | | | | | 0 | Divider (selection at input) |
| 8 | −1 | | | | | | | 1 | −1 | −1 | | | | −2 | Divider (selection at output) |
| 9 | | | | | | | | | 1 | | −1 | | | 0 | Storage |
| 10 | | | | | | | | | | 1 | −1 | | | 0 | Storage |
| 11 | | | | | | | | | | | 1 | −1 | | 0 | Divider (selection at input) |
| 12 | −1 | | | | | | | | | | | 1 | −1 | −1 | Divider (selection at output) |
| 13 | | | −1 | | | −1 | | | | | | | 1 | −1 | Divider (selection at output) |

An exemplary matrix $A(G)$ of the structure depicted in Figure 3 is presented in Table 1.

### 2.6. Projection of the Structural Model into a Set of Relational Models

The proposed structural model accounts for all possible relationships between system objects, but not all relationships can occur at the same time. The above applies to objects in the divider class which can occupy only one of the two possible states (at any given moment, a divider can have only one active output, and the valve can be open or closed). The above implies that the structural model is a set of all possible relational models

$$M_S = M_{R1} \cup M_{R2} \cup \ldots \cup M_{Ri} \cup \ldots \cup M_{RL_R} \tag{5}$$

and the number of models $L_R$ is a product of the number of states of all dividers class objects in the modeled structure

$$L_R = \prod_{i=1}^{n} s_i \tag{6}$$

where $n$—number of objects, $s_i$—number of states of the $i$-th object. A total of 384 relational models can be derived from the tested structure (Figure 3).

A structural model can be projected into a set of relational models with the use of breadth-first search and depth-first search algorithms that are widely applied in general graph theory [44,45]. However, these algorithms are characterized by considerable space and time complexity (which is proportional to the total number of vertices and edges in the searched graph); therefore, a numerical projection algorithm was developed based on the following scheme:

1. Calculate the number of relational models $L_R$;
2. Project the structural model into a set of relational models;
3. Review the set of objects and identify successive objects in the dividers class;
4. Define the number of states $s_i$ for every $i$-th object in the dividers class;
5. Divide the set of relational models into $L_i$ subsets, where $L_i = L_R$ at $n = i$;
6. In each subsequent subset, in relational models belonging to a given subset, leave only one output of the $i$-th object that is related to its subsequent state, and eliminate the remaining outputs.

The results of the projections of the tested structural model into a set of relational models are presented in Figure 5. The analyzed example indicates that the generated relational models may contain objects that do not participate in the technological process. This group of objects includes objects that satisfy the first formal constraint (Section 2.5) including storage class objects (s1 and s2) in model T0000001, as well as objects that do not satisfy the second and third formal constraints, including divider class objects d2 and d8, and a separator class object (cl.) in model T0000315. These objects are inactive, and they should be eliminated from the relational model.

### 2.7. Elimination of Isomorphic Relational Models

When inactive objects (described in Section 2.6) are eliminated from the generated relational models, certain groups of models will have the same structure; therefore, only one model from every such group can be included in the analysis. Each relational model can be represented by a graph (Section 2.4); therefore, it can be assumed that models from the same group are represented by isomorphic graphs. Two graphs can be considered isomorphic if the bisection of one graph is a partition of its vertex set into the vertex set of the other graph [46]. Two graphs are isomorphic if the size of vertex sets and arc sets is equal in both graphs [47]. In general, the isomorphism of two graphs is not easy to verify, and there are no universal algorithms for resolving this issue. However, this problem can be easily overcome in the analyzed structure because every graph vertex representing a specific object is assigned the same number in all models, and these objects have a fixed sequence in the incidence matrix. The implemented numerical algorithm for identifying isomorphic model comprises three steps:

1.  Compare the number of vertices in two graphs (the number of objects in two relational models)—if both graphs have the same number of vertices, proceed to step 2;
2.  Compare both lists of vertices—if they contain vertices with the same numbers, and if vertices have the same sequence, proceed to step 3;
3.  Compare incidence matrices—if the matrices are identical, mark the second relational model as isomorphic with the first model and remove it from the list of models.

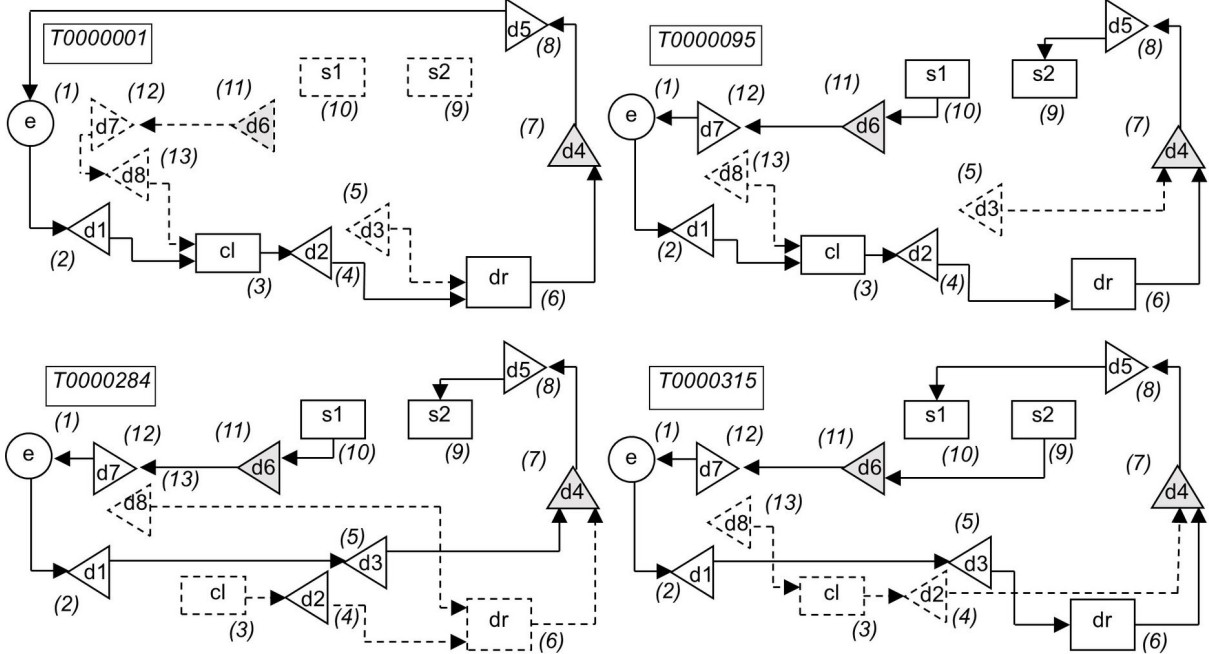

**Figure 5.** Projections of a general structural model (Figure 3) into a set of relational models.

Seven models that are isomorphic with the structure of model T0000001 (Figure 5) have been identified in the general set of relational models (Figure 6). When isomorphic models are eliminated, the size of the set of relational models is reduced from 384 to only 138 models.

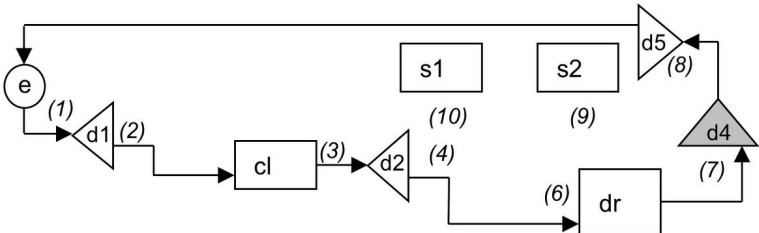

**Figure 6.** Active structure of models T0000001, T0000002, T0000003, T0000004, T0000049, T0000050, T0000051 and T0000052.

## 2.8. Simplification of Relational Models and Graph Homeomorphism

Homeomorphism is an equivalence relationship in a set of finite connected compact graphs [48]. A graph that is homeomorphic to the source graph is intuitively derived by inserting a new vertex in a chain of edges or, conversely, by removing a vertex and replacing edges that are incidental to that vertex with an edge connecting the previous vertex with the next vertex in the chain. This procedure can be applied to simplify a model's structure. The possibility of removing a given vertex and the represented object from the modeled structure is not determined by formal reasons, but by the generality of the model and the objective of modeling. In the modeled structure, it can be generally assumed that

a vertex can be removed if it represents an object of a relational model that satisfies the following criteria:

1. The parameters describing the state of the streams (Section 2.2) at the input and output of the *i*-th object do not change, or the changes do not significantly impact the objective of modeling;
2. The device represented by the modeled object is not a bottleneck, i.e., its maximal throughput cannot be lower than the throughput of the next device in the process line;
3. If one of the modeling objectives is to determine energy consumption in the technological process, only devices whose energy consumption does not significantly affect the overall energy balance can be omitted.

It can be assumed that all dividers class objects representing valves and separators in a real-world process line generally meet the above criteria because

- They do not perform any subprocesses;
- They should not limit the throughput of other devices in a well-designed line;
- These devices are passive, have minimum energy requirements, and consume energy only during changes in state.

In this stage of transformation, graph structures contain only elementary and simple chains; therefore, vertices and edges do not occur more than once in the chain. The elimination of vertices representing dividers class objects simplifies the structure of the graph (Figure 7), but does not reduce the number of graphs; therefore, the size of the set of relational models (Section 2.7) remains unchanged. The above also applies when vertices representing transport class objects are removed, provided that they meet the above criteria.

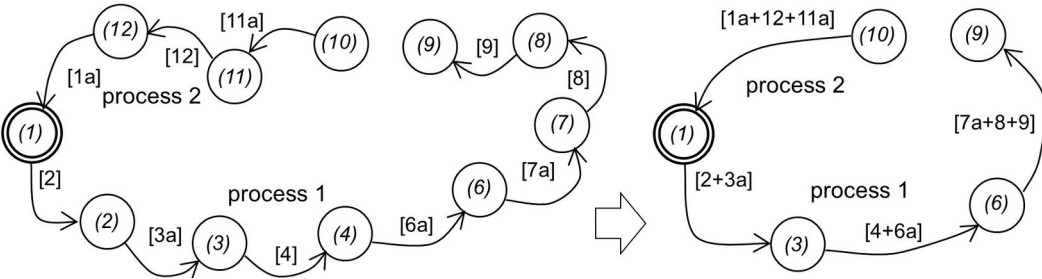

**Figure 7.** Simplification of a relational model (example).

### 2.9. Simultaneous Processes and Graph Connectivity

In graph theory, a directed graph is strongly connected if there is a path between any ordered pair of its vertices [49]. After projection (Section 2.6) and simplification (Section 2.8), the resulting graphs of relational models can belong to one of the two categories: strongly connected graphs or disconnected graphs. Strongly connected graphs represent only simple models or models with a high degree of simplification. These graphs are also Euler graphs and Hamiltonian graphs [50], which implies that the path traverses each arc and each edge of the graph exactly once. At a given moment, a processing line configured based on this model can perform only one process that is an ordered sequence of successive subprocesses involving all active devices in that line.

However, the vast majority of the generated relational models belong to the group of disconnected graphs (Figure 7), where several independent paths and independent vertices can be identified. Each path forms a strongly connected subgraph of the original graph (hypergraph), and it represents a group of subprocesses. In this configuration, several mutually independent processes are performed simultaneously by the processing line. In turn, independent vertices represent stationary subprocesses that are performed only by a single device. These subprocesses include grain storage in a silo and grain drying in a batch dryer.

### 2.10. Functionally Incorrect Processes

For the sake of simplicity, relational models in all groups of graphs have to be subjected to another formal analysis to eliminate models that contain chains describing looped, incomplete and meaningless processes. A process is looped when the corresponding subgraph contains vertex $V(i)$ with one adjacent outgoing arc and at least two adjacent incoming arcs, and does not contain vertex $V(j)$ that is incidental only to the outgoing arc.

$$\begin{cases} \underset{i}{\exists}[d^-(i) = |V^-(i)| > 1 \wedge d^+(i) = |V^+(i)| = 1] \\ \neg\underset{j}{\exists}[d^-(j) = |V^-(j)| > 0 \wedge d^+(j) = |V^+(j)| = 0] \end{cases} \tag{7}$$

The types of objects represented by chain vertices have to be additionally analyzed to identify the two remaining groups of processes. In an incomplete process, the last vertex of the chain represents an object that does not belong to the storage or environment-output class, whereas in a meaningless process, the chain does not contain any vertices representing objects of separator, dryer or storage classes.

The relational models derived from the general structural model are formally validated, and isomorphic models, homomorphic models and models containing chains that describe looped, incomplete and meaningless processes are removed to reduce the overall number of models that can be considered in successive stages of modeling. In the analyzed structure, the set of 384 relational models was reduced to 65 models (246 isomorphic and homomorphic models, 30 models with looped processes, 32 models with incomplete processes, and 11 models with meaningless processes were eliminated).

### 2.11. Connected Processes and Degree of the Subgraph of a Relational Model

Connections can exist between the main processes represented by objects that are linked by the grain stream, as well as between a main process and an auxiliary process (such as supply of the air stream to the dryer and evacuation of the water stream in a stream of air). These processes can be identified by analyzing the degree of the subgraph and the degrees of the identified vertices [51]. The connections between the main processes are described by chains in a common graph which has the following characteristics:

1. There exists more than one node with indegree 0 and outdegree 1 (only the outgoing arc is incident to the vertex);
2. There exists exactly one node with outdegree 0 (only the incoming arc is incident to the vertex);
3. The indegree of the graph is higher than 1 (there exists at least one vertex with indegree higher than 1 and outdegree equal to 1);
4. These criteria are met by the first three layers of the incidence matrix (Section 2.4), i.e., objects connected by grain, impurities and water streams.

$$\begin{cases} \underset{i}{\exists}[d^-(i) = |V_k^-(i)| = 1 \wedge d^+(i) = |V_k^+(i)| = 0] \\ \underset{j}{\exists}[d^-(j) = |V_k^-(j)| = 0 \wedge d^+(j) = |V_k^+(j)| = 1] \\ \Delta(G_k^+) > 1; \quad ||i|| > 1; \quad ||j|| = 1; \quad k = [1,2,3] \end{cases} \tag{8}$$

A main process is connected with an auxiliary process during grain cleaning, drying and active aeration. The chains describing these processes are connected and then disconnected at the identified vertices of graphs representing separator or dryer class objects. A stream of air is additionally applied in all objects, excluding the mechanical separator.

In a given moment, a specific task is assigned to each device that actively participates in the technological process. The type and number of possible tasks is determined by an object's class. The task performed by an object in a given process can be identified by analyzing the incidence of the corresponding vertex to the arcs representing each of the four classes of streams.

### 3. Exemplary Model of a Grain Drying Line

An exemplary model was developed based on the structure of a real-world grain drying line. A general diagram of the modeled line is presented in Figure 8.

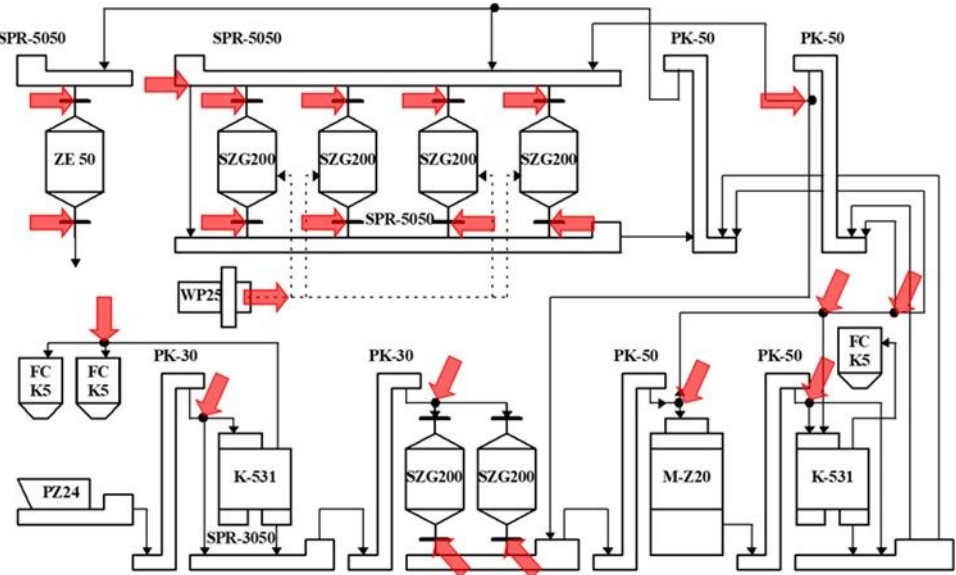

**Figure 8.** A general diagram of a real-world grain drying line.

Grain is transported to a receiving hopper, and it undergoes preliminary cleaning, drying and final cleaning in the line. Processed grain is stored in four silos, and it is transported to truck loading silos before dispatch. Grain is transported in the facility by bucket elevators and redler chain conveyors. The line is configured and grain flow is controlled by valves and separators representing objects of the dividers class (red arrows in Figure 8). The directed graph of the modeled structure is presented in Figure 9.

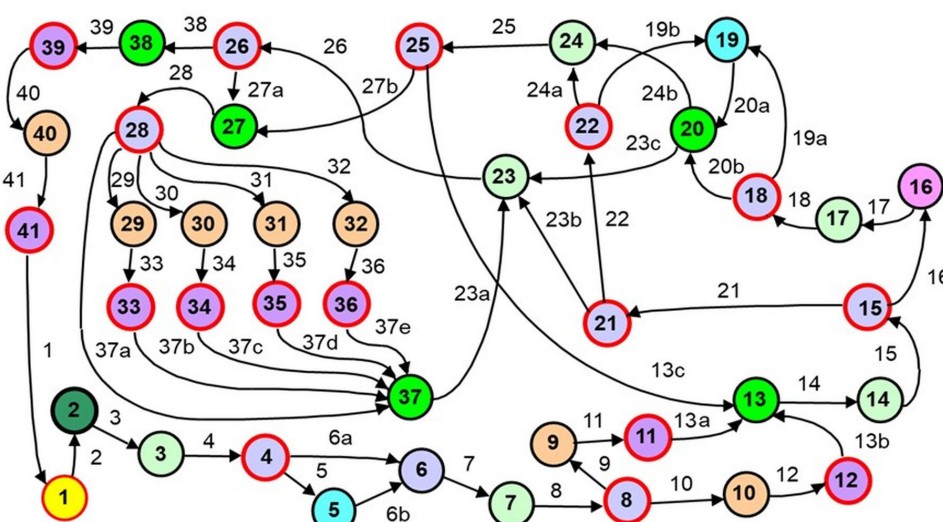

**Figure 9.** Graph presenting the structure of a real-world processing line (1—environment; 2—receiving hopper; 3, 7, 14, 17, 23, 24—bucket elevator; 13, 20, 27, 37, 38—redler chain conveyor; 5, 19—grain cleaner; 16—dryer; 9, 10, 29, 30, 31, 32, 40—storage (silo); 11, 12, 33, 34, 35, 36, 39, 41—valve; 4, 6, 8, 15, 18, 21, 22, 25, 26, 28—separator).

By projecting the graph (Section 2.7), a total of 1,638,400 relational models can be generated from the modeled structure when all possible states of the modeled objects are

taken into account (Equation (6)). When isomorphic models are eliminated (Section 2.7), the set is reduced to 159,153 models (1,307,077 models are isomorphic, 177,400 models describe looped processes, 15,437 models describe incomplete processes, and 6656 models describe meaningless processes), which represents the number of unique configurations that can be obtained by changing separator and valve settings. However, the vast majority of these configurations are described by disconnected graphs (Section 2.9) where several independent paths and independent vertices can be identified. A graph representing one of the numerous projections (T0140028) of the general structure (Figure 8) is a disconnected graph, and it is presented in Figure 10. The relational model represented by that graph supports the simultaneous performance of five processes:

1.　Grain loading (1), preliminary cleaning (5) and transport to a silo (9);
2.　Grain unloading from a silo (10), drying (16) and transport to a silo (30);
3.　Grain unloading from a silo (31) and transport to a facility outside the processing line (1);
4.　Long-term storage of grain in a silo (32);
5.　Active aeration of grain (43) in a silo (33).

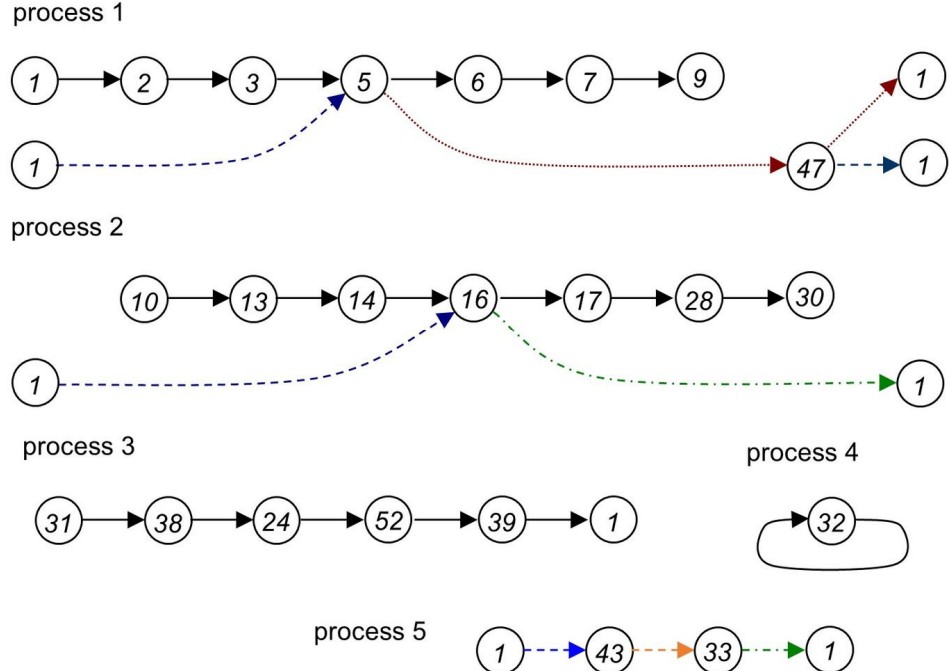

**Figure 10.** Exemplary structure of a disconnected graph that supports the simultaneous performance of many processes.

However, some configurations describe formally incorrect processes (Section 2.10), including 177,400 looped processes, 15,437 incomplete processes and 6656 meaningless processes. Examples of such processes are presented in Figure 11.

In turn, an analysis of the degree of the subgraph of relational models and the degrees of the identified vertices (Section 2.11) indicates that some of the generated models describe connected processes. Figure 12 presents a graph of the main connected processes (T189427) for the line shown in Figure 9. In this case, grain is simultaneously unloaded from silos (10 and 41) and transported to a facility outside the processing line (1).

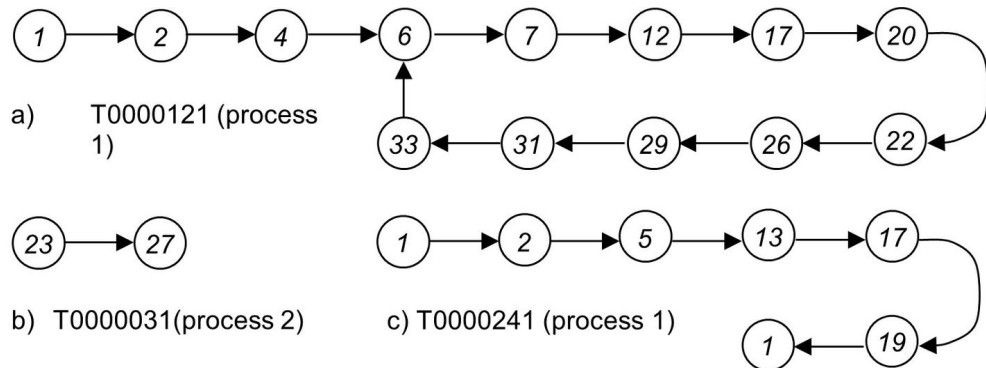

**Figure 11.** Graphs representing processes that do not satisfy formal requirements: (**a**) looped process, (**b**) incomplete process, (**c**) meaningless process.

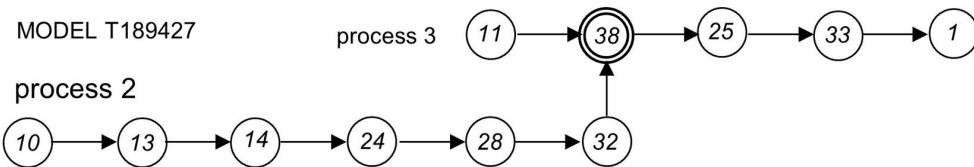

**Figure 12.** Exemplary graph of connected processes.

## 4. Grain Drying Line—Implementation

The presented method represents a preliminary stage in the process of developing a control system for decision-making support in an industrial grain drying line. The control system performs control, measurement and regulatory tasks (Figure 13). An effective control system should support the optimization and partial automation of a process, and cost minimization to achieve other priority goals in the adopted strategy (output maximization, end-products with desirable attributes) can be the optimization criterion.

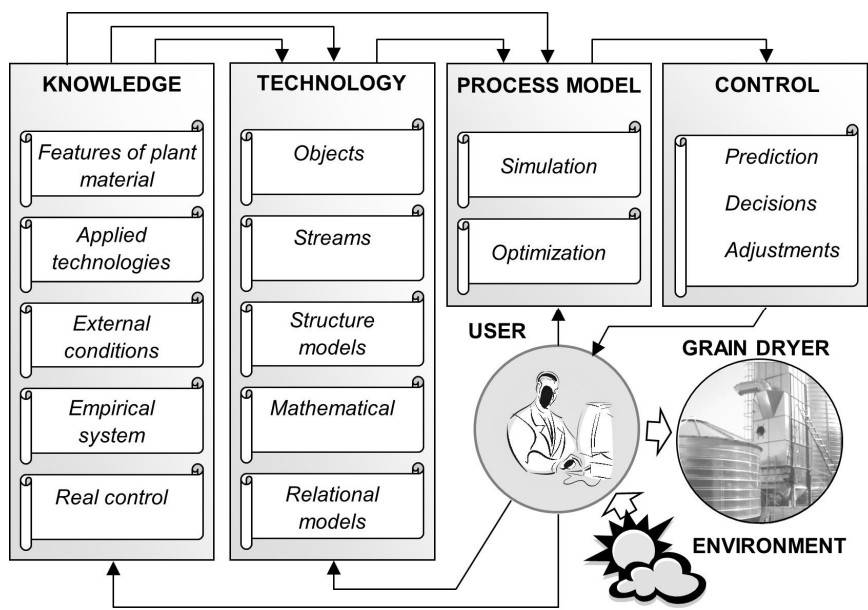

**Figure 13.** Functional diagram of the control system.

The system was implemented in the database environment with the use of object modeling techniques. The tasks to be performed by the defined drying line can be selected or a new line can be entered in the main user interface window (Figure 14). The data required for the simulation can be entered, edited, and processed with the use of the available options.

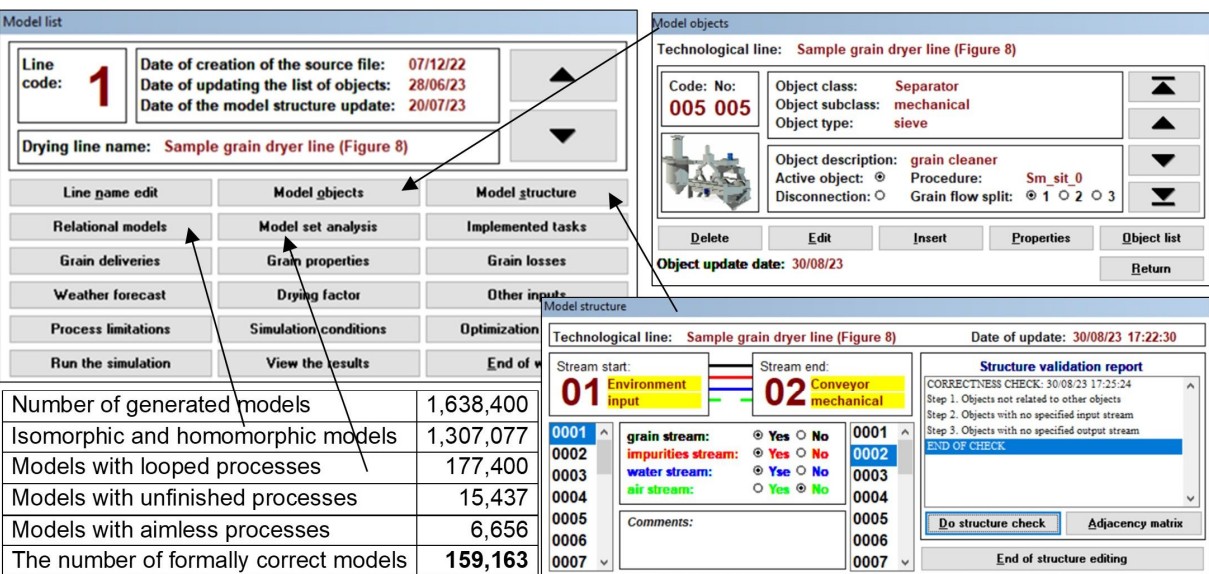

| Number of generated models | 1,638,400 |
|---|---|
| Isomorphic and homomorphic models | 1,307,077 |
| Models with looped processes | 177,400 |
| Models with unfinished processes | 15,437 |
| Models with aimless processes | 6,656 |
| The number of formally correct models | **159,163** |

**Figure 14.** User interface (selected features).

"Model objects" is a set of options for editing the list of objects in the modeled system and the properties of a selected object. The structure of the modeled line can be saved in the dialogue box in the "Model structure" option. Line structure is defined by connecting the inputs and outputs of successive objects with the use of grain dry matter streams, impurities streams, water streams, and air streams. The type of connected objects and the correctness of the defined connections are analyzed automatically during the operation. Formally incorrect connections are blocked by the system, and the relevant information is communicated to the user in the "Notes" section. The formal correctness of the modeled structure can also be verified by the user. After the operation, a report is displayed on the right side of the dialogue box. The defined structure is always verified at the end of the operation, and the result is stored in the system. Only formally correct structures can be used in successive stages of modeling. Experienced users can also edit the structure of the grain drying line at the level of the binary adjacency matrix.

"Relational models" is a group of options for creating a set of relational models of all possible methods of connecting devices into functional processing lines. Each operation has to be conducted in a specified order, as previously described. The end of each operation is marked in time. If the operations are not processed in a chronological order, the system will ask the user to repeat the omitted operations. The user can proceed to the next modeling step only if all operations have been successfully completed.

In the "Model set analysis" stage, models that describe technologically incorrect processes, identify models with connected processes and auxiliary processes are eliminated from the set of relational models. Successive operations have to be performed in a preset order, and the list of processes should be updated in the last step.

## 5. Conclusions

The devices that make up a grain drying line and participate in the drying process have a chain structure, where the grain stream at the output of one device constitutes the input stream of the next device in the chain. This process can be analyzed as a sequence of partial subprocesses that are bound by asymmetric relations. Each subprocess is linked with a specific device, and some devices can perform several tasks sequentially or simultaneously.

The results of the conducted tests indicate that the proposed method fulfills the described objectives and can be applied to model technological processes in agriculture based on the specificity of the modeled processes, the relevant decisions and external conditions. Elements of graph theory were used to develop an effective method for modeling a complex system and to formally validate the model's structure. The proposed algorithms for

projecting a general structural model and evaluating the functionality of the generated set of relational models support complex analyses of multiple process variants. The presented method can be also adapted for use in other areas, such as processing lines in the agri-food industry.

**Author Contributions:** Conceptualization, R.M.; methodology, R.M.; software, R.M.; validation, R.M.; formal analysis, R.M.; investigation, R.M.; data curation, R.M.; resources, R.M.; writing—original draft preparation, R.M., E.J. and M.M.; writing—review and editing, R.M., E.J. and M.M.; visualization, R.M. and E.J.; supervision, R.M., E.J. and M.M.; project administration, R.M., E.J. and M.M.; funding acquisition, R.M., E.J. and M.M. All authors have read and agreed to the published version of the manuscript.

**Funding:** This research received no external funding.

**Data Availability Statement:** The data presented in this study are available on request from the corresponding author.

**Conflicts of Interest:** The authors declare no conflict of interest.

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
