# Peer review of "The Use of Graph Theory for Modeling and Analyzing the Structure of a Complex System, with the Example of an Industrial Grain Drying Line"

_processes, doi:10.3390/pr11102812_

Round 1

Reviewer 1 Report

The authors try to use graph theory to develop models for industrial processes, they give an industrial grain drying line as an example. This research is interesting and promising. My concerns are listed in the following.

  1. In order to construct the graph model, you use a simplified hypothetical grain drying line. For a real industrial process, will it be too complicated to construct the directed graph? There are many loops and objects in practical production.
  2. When you want to apply the graph theory to other industrial processes, do you have to construct the graph model from the beginning?
  3. What is the actual benefit and future application of your model based on graph theory and relational model?
  4. What is the limitation and difficulty in applying graph theory to model industrial process?
  5. In line 245, “Fig. 3” should be Figure 3.

Author Response

Response to Reviewer 1 Comments

General comment: The authors try to use graph theory to develop models for industrial processes, they give an industrial grain drying line as an example. This research is interesting and promising. My concerns are listed in the following.

Response: Thank you for an insightful analysis of our manuscript and valuable comments. To address the Reviewer's concerns, we have added a new paragraph to the Introduction section and we have introduced a new section entitled "Grain drying line – implementation". The purpose of this section was to describe the performed operations in a broader context and to implement them in a programming environment with the use of object modeling techniques and general graph theory.

Comment #1: In order to construct the graph model, you use a simplified hypothetical grain drying line. For a real industrial process, will it be too complicated to construct the directed graph? There are many loops and objects in practical production.

Response: Indeed, a simplified grain drying line was used to construct the graph model. However, this line has the functionality of a real-world industrial process. The simplified grain drying line was introduced to preserve the clarity of the presentation. An exemplary model of a grain drying line was presented in section 3 and the new section 4. In the proposed method, the number of object classes does not affect the structural complexity of the graph. The user is only required to define a list of objects and connections ("Model objects" and "Model structure" options in Figure 14), whereas the graph is generated automatically and stored as an incidence matrix. 

Comment #2: When you want to apply the graph theory to other industrial processes, do you have to construct the graph model from the beginning?

Response: Significant differences were observed in an earlier stage of developing the object model, which is why the classes of objects and streams had to be defined, and a list of attributes and the applied methods had to be generated. However, the modeling procedure does not have to be modified at the stage of building a general graph model.

Comment #3: What is the actual benefit and future application of your model based on graph theory and relational model?

Response: The main strength of the proposed model is its universal applicability. Elements of the graph theory and relational databases can be applied to develop an effective tool for modeling complex systems and validating the formal correctness of their structure. The proposed algorithms for projecting a general structural model and evaluating the functionality of the generated set of relational models support complex analyses of multiple process variants. For example, as a result of automated formal processes, the generated set of 1,638,400 relational models (Figure 14) was reduced to  159,153 models (9.7%), which significantly speeds up calculations during simulations and process optimization.

Comment #4: What is the limitation and difficulty in applying graph theory to model industrial process?

Response: The computational complexity of the algorithm for eliminating isomorphic graphs is the main limitation of the proposed method. In general, the isomorphism of two graphs is not easy to verify, and there are no universal algorithms for resolving this issue. In our study, this problem was addressed by comparing the incidence matrices. However, if the divide and conquer paradigm is used to design algorithms for more complex structures, this task can be recursively split into a larger number of smaller subproblems.

Comment #5: In line 245, “Fig. 3” should be Figure 3.

Response: The relevant correction was made.

Reviewer 2 Report

The abstract section is notably brief and overly generalized, lacking the presentation of concrete data, particularly with regard to the drying process, which is a key aspect discussed in the article. Predominantly, the paper is centered around detailing the method employed, leaving the results with inadequate explanation, thereby impeding the overall readability of the text.

The manuscript under consideration holds thematic significance and relevance for researchers and professionals engaged in the respective field.

The distinctiveness of this work is not readily apparent to me.

With certain adjustments and enhanced elucidation provided by the authors, the manuscript could be deemed suitable for publication in the Processes journal.

The title effectively encapsulates the manuscript's content, while the arrangement and structure of the document maintain appropriateness.

The subject matter aligns well with the journal's scope, and the paper's objective is effectively delineated.

It is crucial to undertake a comprehensive evaluation of analogous works in this domain, extending beyond a localized perspective. This examination should spotlight their inadequacies, which can subsequently be formulated into the foundational theses of this paper.

The results and discussions demonstrate a reasonably strong correlation with the referenced literature. While the literature review is comprehensive and adequately executed, an infusion of more recent references would be beneficial.

Furthermore, the originality of the research could be expounded upon more explicitly.

In the section labeled "Conclusions", please offer a succinct summary of the principal innovative scientific and research accomplishments. Additionally, furnish a more intricate roadmap for future investigations within the subject domain of the paper.

On the whole, the language and presentation are fitting, and the research methodology is cogently expounded upon, enabling replication by other scholars. The progression of results is logically coherent, with a well-developed analysis.

However, the predominant limitation lies in discerning the novelty quotient of this paper against the backdrop of existing research by peers.

The authors are encouraged to invest extra effort into substantiating how this work constitutes a substantive contribution to the research landscape.

To conclude, the proposed manuscript exhibits potential for publication in the Processes journal, contingent on requisite adjustments and a more pronounced exhibition of novelty and contributions to the field.

Author Response

Response to Reviewer 2 Comments

General comment: The abstract section is notably brief and overly generalized, lacking the presentation of concrete data, particularly with regard to the drying process, which is a key aspect discussed in the article. Predominantly, the paper is centered around detailing the method employed, leaving the results with inadequate explanation, thereby impeding the overall readability of the text.

Response: Thank you for an insightful analysis of our manuscript and valuable comments. To address the Reviewer's concerns, we have added a new paragraph to the Introduction section, and we have introduced a new section entitled "Grain drying line – implementation". The purpose of this section was to describe the performed operations in a broader context and to implement them in a programming environment with the use of object modeling techniques and general graph theory. The presented method represents a preliminary stage in the process of developing a control system for decision-making support in an industrial grain drying. Algorithms for projecting the general structural model and evaluating the functionality of the obtained relational models are developed in this stage. These algorithms support complex analyses of multiple process variants. For example, as a result of automated formal processes, the generated set of 1,638,400 relational models (Figure 14) was reduced to  159,153 models (9.7%).

We agree with the Reviewer that grain drying is the main subprocess. However, grain drying involves other subprocesses, including transport, separation, and storage, as well as decisions concerning the line's structure and state, the designation of raw materials, and the parameters of each device. Each of these subprocesses requires an appropriate mathematical model (Figure 13). In the proposed solution, these models are implemented as methods that are assigned to specific classes of the object model. A comprehensive description of such models remains outside the scope of this paper. A model for grain drying subprocesses has been previously presented by Myhan, R., Markowski, M., Generalized Mathematical Model of the Grain Drying Process. Processes 2022, 10, 2749. https://doi.org/10.3390/pr10122749.

Comment #1: The manuscript under consideration holds thematic significance and relevance for researchers and professionals engaged in the respective field.

Response: Thank you for the positive feedback.

Comment #2: The distinctiveness of this work is not readily apparent to me.

Response: We have added a new paragraph to the Introduction section, and we have introduced a new section entitled "Grain drying line – implementation" to describe the performed operations in a broader context.

Comment #3: With certain adjustments and enhanced elucidation provided by the authors, the manuscript could be deemed suitable for publication in the Processes journal.

Response: We hope that the revised manuscript is suitable for publication.

Comment #4: The title effectively encapsulates the manuscript's content, while the arrangement and structure of the document maintain appropriateness.

Response: Thank you for the positive feedback.

Comment #5: The subject matter aligns well with the journal's scope, and the paper's objective is effectively delineated.

Response: Thank you for the positive feedback.

Comment #6: It is crucial to undertake a comprehensive evaluation of analogous works in this domain, extending beyond a localized perspective. This examination should spotlight their inadequacies, which can subsequently be formulated into the foundational theses of this paper.

Response: Models of specific production processes and commercial steady-state process simulators (such as ProSimPlu and VMGSim) have been extensively researched, but models of alternative methods of managing raw materials that account for the structure and state of the processing line as well as external conditions have not been proposed in the literature to date. Such an attempt has been made in our previous studies: Myhan, R. Analysis and Modeling of Complex Agricultural Systems and Technological Processes; Rozprawy i Monografie, Uniwersytet WarmiÅ„sko-Mazurski w Olsztynie: Olsztyn, 2009 (in Polish), and Jachimczyk, E., Myhan, R. Generation of alternative methods for managing raw materials to support decision-making in the dairy industry; Food and Bioproducts Processing. 133 (2022) 140–152, https://doi.org/10.1016/j.fbp.2022.04.001. In the revised manuscript, we have added a new paragraph to the Introduction section to address the above concerns.

Comment #7: The results and discussions demonstrate a reasonably strong correlation with the referenced literature. While the literature review is comprehensive and adequately executed, an infusion of more recent references would be beneficial. Furthermore, the originality of the research could be expounded upon more explicitly.

Response: Please refer to the explanation provided in Response #6.

Comment #8: In the section labeled "Conclusions", please offer a succinct summary of the principal innovative scientific and research accomplishments. Additionally, furnish a more intricate roadmap for future investigations within the subject domain of the paper.

Response: The manuscript was revised to address the Reviewer's suggestions. A succinct summary of the performed operations was provided in a new section (4. Grain drying line – implementation).

Comment #9: On the whole, the language and presentation are fitting, and the research methodology is cogently expounded upon, enabling replication by other scholars. The progression of results is logically coherent, with a well-developed analysis.

Response: Thank you for the positive feedback.

Comment #10: However, the predominant limitation lies in discerning the novelty quotient of this paper against the backdrop of existing research by peers. The authors are encouraged to invest extra effort into substantiating how this work constitutes a substantive contribution to the research landscape.

Response: Please refer to the explanation provided in Response #8.

Comment #11: To conclude, the proposed manuscript exhibits potential for publication in the Processes journal, contingent on requisite adjustments and a more pronounced exhibition of novelty and contributions to the field.

Response: Once again, thank you for the careful perusal of our manuscript and valuable suggestions. We hope that the revised manuscript is suitable for publication.

Reviewer 3 Report

The manuscript describes a method for analyzing and modeling a complex agrotechnological system on the example of an industrial grain drying line. Elements of graph theory are used, which my report focuses on because this is my speciality.

Overall, the manuscript is readable, but I can not judge the relevance of the agrotechnological considerations. At the very low level of my expertise, it seems to me that this research may be important for improved modeling of practical processes. 

Unfortunately, considering the details where the application of graph theory is used, there are many formal inconsistencies, explained below. In fact, the definitions are confusing to extend that I have no idea what is really meant, so I can not imagine how it can be used. 

Therefore, I can not recommend publication of the manuscript in its present form. 

(Some) of the problems (section 2.4):

- set of vertices  is written as <o_1, o_2, ..., o_n>, which is unusual notation  (in mathematics, usual notation would be {o_1, o_2, ..., o_n})

- set of branches (usual term in graph theory is edges, directed edges or arcs) is written as <S_m, S_z, S_w, S_p> which is confusing. Is this a set of 4 elements ? I guess you want to say that S is a UNION of four subsets  S = S_m \cup S_z \cup S_w \cup S_p  (\cup is LaTeX command for the union symbol U)

- furthermore, perhaps (I am guessing again) you want to say that S_m, S_z, S_w, and S_p are PAIRWISE disjoint, i.e S_i \cap S_j = \emptyset for all pairs i\not=j.

- page 6, line 178-182: what is "class"? (I guess subsets S_* are called classes of branches?)

- page 6, line 190-...: requirements are unclear, starting with the first one, where the statement obivously is wrong (such that WHAT ?)

- and so on

Author Response

Response to Reviewer 3 Comments

General comment: The manuscript describes a method for analyzing and modeling a complex agrotechnological system on the example of an industrial grain drying line. Elements of graph theory are used, which my report focuses on because this is my speciality. Overall, the manuscript is readable, but I can not judge the relevance of the agrotechnological considerations. At the very low level of my expertise, it seems to me that this research may be important for improved modeling of practical processes. Unfortunately, considering the details where the application of graph theory is used, there are many formal inconsistencies, explained below. In fact, the definitions are confusing to extend that I have no idea what is really meant, so I can not imagine how it can be used.

Response: We appreciate the reviewer's insight in evaluating the formal side of our work. At the outset, we would like to point out that we do not feel ourselves to be specialists in graph theory. The presented method is a preliminary stage of activities aimed at developing a control system to support the operation of grain dryer process lines (new Chapter 4). At this stage, our proposal was based on elements of object-oriented modeling and general graph theory. Two groups of classes are defined in the object model. Instances of classes from the first group represent elements of line equipment, and instances of classes from the second group represent streams of material agent that move between instances of object classes from the first group. In the "graph model" instances of classes from the first group are represented by vertices of the graph, and instances of classes from the second group are directed edges. The use of the "graph model" has made it possible to develop an effective tool for modeling such a system and formally analyzing the correctness of writing its structure. The proposed method makes it possible to transform the general structure model into a set of relational models of the system, formally evaluate the functionality of the resulting models, and comprehensively analyze possible variants of process implementation.

Like the Reviewer, we believe that in addition to the substantive content, the notation used to convey it is also important. The problem is that many notations can be found in the literature [39, 40, 41, 42, 45, 46, 47, 48, 49]. In our case, we tried to keep the notation: Wilson, R.J. Introduction to Graph Theory, 5th ed.; Pearson: London, 2010. Nevertheless, following the reviewer's comments, we propose the following changes:

Comment #1: - set of vertices  is written as <o_1, o_2, ..., o_n>, which is unusual notation  (in mathematics, usual notation would be {o_1, o_2, ..., o_n}).

Response: Corrected.

Comment #2: - set of branches (usual term in graph theory is edges, directed edges or arcs) is written as <S_m, S_z, S_w, S_p> which is confusing. Is this a set of 4 elements ? I guess you want to say that S is a UNION of four subsets  S = S_m \cup S_z \cup S_w \cup S_p  (\cup is LaTeX command for the union symbol U).

Response: Corrected.

Comment #3: - furthermore, perhaps (I am guessing again) you want to say that S_m, S_z, S_w, and S_p are PAIRWISE disjoint, i.e S_i \cap S_j = \emptyset for all pairs i\not=j.

Response: Sm, Sz, Sw i Sp are sets of instances of object model stream classes.

Comment #4: - page 6, line 178-182: what is "class"? (I guess subsets S_* are called classes of branches?).

Response: In graph notation, the stream class is represented by the index „k” (a(i,j,k)).

Comment #5: - page 6, line 190-...: requirements are unclear, starting with the first one, where the statement obivously is wrong (such that WHAT ?) and so on.

Response: In our opinion, the defined conditions are correct (they were also verified at a later stage of the simulation). Note that four graphs are analyzed (one for each class of streams) stored in one four-layer incident matrix (eq. 4), where: i – vertex index, j – edge index, k – stream class index.

Reviewer 4 Report

This paper applies standard Graph Theory methodology to model a Grain drying Process. No extensions of Graph theory Methodology is presented. The application on Grain Drying process is relevant and useful for agro-food processing industry. The model presented is plausable, however no numerical or experimental results or verifications are presented.

A discussion of specific applications with numerical results will be useful.

Author Response

Response to Reviewer 4 Comments

General comment: This paper applies standard Graph Theory methodology to model a Grain drying Process. No extensions of Graph theory Methodology is presented. The application on Grain Drying process is relevant and useful for agro-food processing industry. The model presented is plausible, however no numerical or experimental results or verifications are presented.

A discussion of specific applications with numerical results will be useful.

Response: We are grateful for the Reviewer's remarks and suggestions. However, we feel that several comments require additional explanations:

  1. This study was not undertaken to model grain drying processes. A model of a grain drying process was presented in our previous paper (Myhan, R., Markowski, M., Generalized Mathematical Model of the Grain Drying Process. Processes 2022, 10, 2749. https://doi.org/10.3390/pr10122749).
  2. Please note that grain drying involves other subprocesses, including transport, separation, and storage, as well as decisions concerning the line's structure and state, the designation of raw materials, and the parameters of each device.
  3. The presented method represents a preliminary stage in the process of developing a control system for decision-making support in an industrial grain drying line. The decision-making process involves the selection of the optimal structure of a grain drying line based on external conditions and the current state of line devices.
  4. The entire system (beginning from a model of the line's structure and ending in process simulation and the optimization of decision-making) was implemented in the database environment with the use of object modeling techniques.
  5. It was not our intention to present an extension of the graph theory methodology, although such attempts were made when designing the algorithm for eliminating isomorphic models. The proposed method can be used to transform a general structural model into a set of relational models, to formally evaluate the resulting models’ functionality, and to comprehensively analyze different variants of the process.
  6. The validation and verification of the proposed system constitutes a separate, but very important issue. In highly complex systems, validation and verification steps have to be performed in successive modeling stages to evaluate the adequacy of the mathematical models applied to different classes of objects, and forecast models of grain delivery and weather conditions. The adequacy of a grain drying model was evaluated in the study cited in point 1.
  7. As regards the Reviewer's claim that numerical or experimental results are not presented in the article, please note that we have a full set of simulation results for the grain drying line presented in Figure 8. However, this dataset is extensive, and it cannot be presented in the manuscript due to volume constrains. The results can be forwarded to a repository indicated by the Editor or forwarded directly to the interested Reviewers.

Round 2

Reviewer 2 Report

The authors have successfully enhanced the overall quality of the manuscript by diligently addressing and incorporating the valuable feedback provided by the reviewers. Their commitment to refining the content based on the constructive comments demonstrates their dedication to producing a high-caliber piece of scholarly work.

Considering the substantial improvements made by the authors in response to the reviewers' input, I strongly recommend that the editor accept the manuscript in its current form. This refined version is well-prepared and possesses the potential to make a meaningful contribution to the academic community, making it a strong candidate for publication in the esteemed Processes journal.

Author Response

Response to Reviewer 2 Comments

General comment: 

The authors have successfully enhanced the overall quality of the manuscript by diligently addressing and incorporating the valuable feedback provided by the reviewers. Their commitment to refining the content based on the constructive comments demonstrates their dedication to producing a high-caliber piece of scholarly work.

Considering the substantial improvements made by the authors in response to the reviewers' input, I strongly recommend that the editor accept the manuscript in its current form. This refined version is well-prepared and possesses the potential to make a meaningful contribution to the academic community, making it a strong candidate for publication in the esteemed Processes journal.

Response:

Thank you very much for taking the time to review this manuscript. Thank you for the positive review and for the good recommendation.

Reviewer 3 Report

The sentence on p6, lines 199-201, is still a logical mess. This means that the authors did not time to seriously consider my remark(s).

I do not wish to read any later versions. 
